# Cerebrospinal Fluid Neurofilaments Light-Chain Differentiate Patients Affected by Alzheimer’s Disease with Different Rate of Progression (RoP): A Preliminary Study

**DOI:** 10.3390/brainsci14100960

**Published:** 2024-09-25

**Authors:** Valeria Blandino, Tiziana Colletti, Paolo Ribisi, Domenico Tarantino, Viviana Mosca, Luisa Agnello, Marcello Ciaccio, Tommaso Piccoli

**Affiliations:** 1Cognitive and Memory Disorders Clinic, AOUP “Paolo Giaccone” University Teaching Hospital, Department of Biomedicine, Neurosciences, and Advanced Diagnostics (Bi.N.D.), University of Palermo, 90127 Palermo, Italy; valeriablandino0@gmail.com (V.B.); tizianacolletti@gmail.com (T.C.); paolo.ribisi6@gmail.com (P.R.); tarantino.domenico95@gmail.com (D.T.);; 2Institute of Clinical Biochemistry, Clinical Molecular Medicine, and Clinical Laboratory Medicine, Department of Biomedicine, Neurosciences, and Advanced Diagnostics (Bi.N.D), University of Palermo, 90127 Palermo, Italy; luisa.agnello@unipa.it (L.A.); marcello.ciaccio@unipa.it (M.C.); 3Department of Laboratory Medicine, University Hospital “P. Giaccone”, 90127 Palermo, Italy

**Keywords:** Alzheimer’s disease, dementia, biomarkers, neurofilaments, CSF, progression

## Abstract

Alzheimer’s disease (AD) is the most common neurodegenerative disorder and a leading cause of dementia. One major challenge for clinicians is accurately assessing the rate of disease progression (RoP) early in the diagnostic process, which is crucial for patient management and clinical trial stratification. This study evaluated the role of cerebrospinal fluid biomarkers—Aβ42, t-Tau, pTau, Neurogranin (Ng), and Neurofilament light-chain (NF-L)—in predicting RoP at the time of AD diagnosis. We included 56 AD patients and monitored cognitive impairment using MMSE scores at diagnosis and during six-month follow-up visits. RoP scores were calculated based on these assessments. Our correlation analyses revealed significant associations between RoP and pTau, Aβ42/Ng ratio, and NF-L levels. When patients were stratified by median RoP values into low-to-moderate (L-M: <2) and upper-moderate (U-M: >2) groups, those in the U-M group had notably higher CSF NF-L levels compared to the L-M group. Logistic regression analysis further demonstrated that elevated CSF NF-L levels were predictive of a faster RoP. These findings highlight the potential of CSF NF-L as a prognostic biomarker for rapid disease progression in AD. By identifying patients at risk for accelerated cognitive decline, CSF NF-L could significantly enhance early intervention strategies and improve patient management in clinical settings.

## 1. Introduction

Alzheimer’s disease (AD) is a highly disabling degenerative disorder of the Central Nervous System (CNS), characterized by a chronic and progressive cognitive deterioration linked to the extracellular deposition of β amyloid (Aβ) in the brain parenchyma and cerebral blood vessels, intraneuronal neurofibrillary tangles, and synaptic dysfunctions, which represent typical disease hallmarks [1,2,3]. AD neuropathology begins years before the appearance of clinical symptoms, so a diagnosis at an early stage of the disease and the possibility to predict the severity and the rate of progression (RoP) represent the main objectives for clinicians in the management of patients with dementia and in their stratification for potential clinical trials testing disease-modifying therapies [4].

The evaluation of AD severity has been based for years on the clinical assessment of cognitive and behavioral impairments, as well as the ability to independently carry out common daily activities [5]. In 2011, Albert and McKhann introduced the use of biomarkers for the diagnosis of the disease in a clinical setting, opening a new era for the management of the disease, especially from the possibility, for the first time, to diagnose AD in patients in the very early stages of the disease (Mild Cognitive Impairment, MCI) [6,7]. At the same time, several publications have led to a change in our view of disease from a purely clinical entity to a clinico-biological one, with the introduction of new terminology and a biomarkers-based classification [8,9,10].

Many authors considered the clinical course of the disease as not linear, with a fluctuating pattern due to endogenous (such as sex, age, comorbidities, and genetic risk factors) or exogenous factors (such as infectious diseases, toxins, and drugs) [11]. For this reason, the need to have a model able to predict RoP at diagnosis emerged for different purposes: better management of affected patients, a correct assignment to different experimental branches in disease-modifying clinical trials, and validation of biomarkers. Doody and colleagues [12] proposed the “pre-progression rate” able to estimate the RoP on the basis of the cognitive assessment by MMSE at the time of diagnosis combined with the diagnostic delay, namely the time that occurred between the symptoms onset and the first clinical evaluation. However, the exact moment of disease onset is not easily identifiable due to possible inaccuracies reported by individuals close to the patient. Recently, novel methods to estimate RoP considered expert disease knowledge, machine learning algorithms, and regression-based prediction models, alone or in combination: thus, this type of approach could increase the accuracy of the patient prognosis [13].

In recent times, the importance of Neurogranin (Ng), a biomarker related to synaptic dysfunction, has emerged in AD. Ng is a postsynaptic protein, particularly expressed in neurons localized in the hippocampus and cerebral cortex, which has a critical role in synaptic function and plasticity [14]. Ng is considered a promising biomarker able to detect synaptic dysfunction in AD patients because its CSF concentrations are increased in AD patients in comparison to healthy subjects and patients affected by other Neurodegenerative disorders (NDDs) [15].

Among different fluid biomarkers, Neurofilaments (NFs) seem to play a crucial role in the processes that lead to faster RoP in NDDs. NFs are heteropolymers that are specific to neuronal cells and belong to the intermediate filament class. Normally, they are highly stable in axons and their turnover is low. When axonal damage or degeneration occurs, they are released in large quantities into the interstitial space, from where they first pass into the cerebrospinal fluid (CSF) and then into the blood [16]. For this reason, NFs are recognized as biomarkers for neurodegeneration [17]. Among the different isoforms, Neurofilament light-chain (NF-L), the most abundant and soluble form detected in biofluids [18], was investigated in a variety of neurological disorders, such as Multiple Sclerosis (MS), AD, frontotemporal dementia (FTD), Amyotrophic Lateral Sclerosis (ALS), atypical parkinsonian disorders (APD), and traumatic brain injury (TBI) [17,19,20,21,22,23,24,25], showing the non-specific nature of axonal damage and its contribution in disease progression. In addition, the blood assay of NF-L is highly sensitive and reproducible, which allows their use for longitudinal studies [19].

In AD, both CSF and plasma NF-L increased in the early stages of the disease and correlated with disease severity [26]. High CSF NF-L levels also correlated with a greater risk of developing MCI in cognitively unimpaired individuals, with faster rates of cognitive decline in MCI and with overall decreased survival in AD dementia, suggesting their role as a prognostic biomarker [27]. In familiar AD, increased levels of NF-L were detected in mutation carriers than in non-carriers, and there was an increase in annual rate in mutation carriers as early as 16 years before the estimated symptom onset, demonstrating their potential role as a biomarker of phenoconversion [28,29]. Other studies found that NF-L could predict clinical conversion from MCI to dementia [22,23,29,30] and that their concentrations were significantly lower in stable MCI patients than in progressive ones [23].

Predicting the RoP of AD is challenging due to its phenotypic heterogeneity [31]. The present study aimed to investigate CSF biomarkers as predictive factors for RoP in AD patients. For this purpose, we analyzed AD core biomarkers (i.e., Aβ 42, Aβ 42/40 ratio, tTau, and pTau), as well as NF-L and Ng, which are widely considered markers of neurodegeneration and synaptic dysfunction, respectively.

## 2. Patients and Methods

### 2.1. Demographic, Clinical, and Neuropsychological Features of Participants

In this retrospective observational study, we selected 56 patients from the Memory Clinic of the Unit of Neurology, University Hospital of Palermo (Italy), diagnosed with Alzheimer’s disease according to the current criteria [6,7]. All patients underwent a complete medical and neurological evaluation, complete routine blood measurement, neuropsychological evaluation, brain MRI and FDG-PET, and CSF withdrawal as routine diagnostic procedures. All recruited patients showed brain atrophy in MRI scans, brain hypometabolism at FDG-PET, and AD core CSF biomarkers abnormalities, being classified as A+T+N+ [8]. APO E genotyping was also determined.

All demographic and clinical information about AD patients was opportunely recorded into a digital database, in which demographic, clinical, neuropsychological, and biological features were reported. Cognitive deficits were measured by the Mini Mental State Examination (MMSE). The details are reported in Appendix A.

For our purposes, we selected patients with at least two MMSE evaluations over time, performed 6 months apart.

We then stratified patients on the basis of their observed rate of progression (RoP), as previously described by Doody et al. [12]. Briefly, we applied the following formula:RoP = (First MMSE–Last MMSE)/time of follow-up (months)

All patients gave their written consent to the procedures, according to the amendments of the Declaration of Helsinki. The study protocol was approved by the Ethics Committee of the University Hospital of Palermo.

### 2.2. Withdrawal, Processing, and Analyses of CSF Samples

CSF withdrawal was performed in polypropylene tubes by lumbar puncture (LP) in fasted patients between 8:00 a.m. and 10:00 a.m. To remove cell debris, CSF samples were centrifuged at 300× *g* for 5 min, aliquoted in propylene tubes, and stored at −80 °C until analysis, according to international consensus protocols [32].

The CSF concentrations of Aβ42, Aβ40, tTau, and pTau were analyzed by chemiluminescence enzyme immunoassay (CLEIA), using the following commercially available kits (Fujirebio Inc. Europe, Gent, Belgium), according to the manufacturer’s instructions: Lumipulse G b Amyloid 1–42 CSF (#230336), Lumipulse G b Amyloid 1–40 CSF (#231524), Lumipulse G pTau 181 CSF (#230367), and Lumipulse G Total Tau CSF (#230312). Commercially available ELISA kits were used to measure CSF levels of Ng (#EQ 6551-9601-L; Euroimmun, Lubeck, Germany) and NF-L (#10-7001 CE; Uman Diagnostic, Umea, Sweden), according to the manufacturer’s instructions [17,33]. We also considered the Aβ42/40 ratio, which showed a significantly better diagnostic performance for AD compared to the CSF Aβ42 concentration [34] and Aβ42/Ng, which we previously described as an index of synaptic dysfunction in AD [33].

### 2.3. Statistical Analyses

All statistical analyses were performed using SIGMAPLOT 12.0 (Systat Software Inc., San Jose, CA, USA) and SPSS version 25 (SPSS Inc., Chicago, IL, USA). To verify the distribution of the data, we used the Shapiro–Wilk method. Normal continuous variables were indicated as mean ± standard deviation (s.d.), while skewed data were included as median with interquartile ranges (IQR). Categorical variables are indicated as frequencies. 

To detect differences between different groups, we performed the Mann–Whitney U-test if data had a non-parametric distribution, or the Student *t*-test if data were normally distributed, also considering effect size. To detect the differences between analyzed groups for categorical variables, we used Chi-square or Fisher’s exact test, as appropriate.

To evaluate any correlation between RoP with demographic, clinical, and biological features of patients, we used Pearson’s analysis for parametric data or Spearman’s analysis, for skewed data.

Receiver operating characteristic curve (ROC) analysis was used to discriminate the ability of selected biomarkers to allocate patients into two different groups according to median values of their RoP, calculating the best cut-off, sensitivity, specificity, the area under curve (AUC) with 95% confidence interval (CI), and the likelihood ratio (LR). AUC was used to identify the discriminatory ability of biomarkers, classified as follows: “excellent” (AUC 0.90–1.00), “good” (AUC 0.80–0.89), “fair” (AUC 0.70–0.79), “poor” (AUC 0.60–0.69) or “fail”/no discriminatory capacity (AUC 0.50–0.59).

Logistic regression analyses were used to estimate the predictive role of CSF biomarkers and RoP, considering values of RoP upper than median values. Moreover, we considered different predictive models derived from the combination of biological variables, and their selection was aided by Akaike Information Criteria (AIC) and Bayesian Information Criteria (BIC). *p* < 0.05 was considered statistically significant.

## 3. Results

### 3.1. Demographic and Clinical Features of Participants

We performed a retrospective observational study on 56 (n = 56) patients affected by AD to explore if their RoP could be predicted on the basis of the levels of CSF biomarkers at the time of diagnosis. For our purposes, we stratified patients into two groups, comparing AD patients with RoP lower than the median value (L-M: <2) to those with RoP upper than the median value (U-M: >2). As shown in Table 1, no significant differences in demographic and clinical features were identified between the two groups.

Spearman’s correlation analyses were used to investigate the association of RoP with the demographic and clinical features of AD patients. No significant correlation was detected (Appendix A). 

### 3.2. CSF Biomarkers in AD Patients Stratified on the Basis of Their RoP

Firstly, the relationship between RoP scores and CSF biomarkers was investigated by Spearman’s correlation analysis. Significant correlations of RoP with pTau (*rho* = 0.283, *p* = 0.034), NF-L (*rho* = 0.432, *p* = 0.001), and Aβ 42/Ng ratio (*rho* = −0.293, *p* = 0.039) were found (Table 2).

Stratifying AD patients into two groups as described above, we found that U-M showed higher CSF concentrations of NF-L in comparison to L-M, while other biomarkers did not reach statistical significance (Table 3, Figure 1).

Subsequently, we investigated the diagnostic accuracy of CSF NF-L to discriminate L-M from U-M. ROC curves showed a significant fair ability of CSF NF-L to discriminate U-M from L-M (A.U.C. = 0.73 C.I.95% (0.590–0.865), *p* = 0.004). The best cut-off for NF-L to distinguish between L-M from U-M, calculated on the basis of Youden’s index, was 671.5 pg/mL, with a sensitivity of 72.7% and a specificity of 70.8% (Figure 2).

### 3.3. Demographic and Clinical Features of Participants and CSF Biomarkers in Predicting RoP of AD Patients

Investigating the contribution of demographic and clinical features of AD patients to predict their RoP, including comorbidities and the use of pharmacological therapies, we did not detect any significant role as potential confounders in predicting RoP (Appendix A). Then, we analyzed the predictive role of CSF biomarkers. For this purpose, we considered the previously described cut-off for Aβ 1–42, Aβ 42/40 ratio, tTau, and pTau to stratify our population on the basis of “pathological” values, i.e., <650 pg/mL, <0.055, >416 pg/mL, and >61 pg/mL, respectively [35]. For Ng and Aβ 42/Ng, we stratified participants according to their median values, which are detailed in Table 2. For NF-L, we subdivided participants according to the cut-off value (i.e., 671.5 pg/mL), as reported above. We found that only CSF levels of NF-L higher than the cut-off value were associated with an increased risk of having a faster RoP (*p* = 0.002; Table 4).

Then, we performed multivariate logistic analyses to verify if NF-L combined with other CSF biomarkers and APO E ε4 genotype is able to better predict RoP in our population. For this reason, we fitted different models of prediction, considering AIC and BIC for the choose of the best possible model. Indeed, we chose NF-L as the “base” in our models because it was the only dependent variable that reached statistical significance in binary logistic regression. As shown in Table 5, the intercept values of all considered models reached statistical significance, with the exception of Models 3 and 4. Then, we provided to model selection, preferring the one with lower AIC and BIC. So, excluding the base model, which showed an AIC of 11.11 and a BIC of 15.16, Model 2, in which NF-L is combined with Ab 42/40, showed the best fit in comparison with the other considered ones. Notably, the combination of NF-L with Ab 42/Ng (Model 7) represented the second best-fitted model, paying attention to the cumulative role of neurodegeneration and synaptic dysfunctions in AD progression. 

## 4. Discussion

Predicting the rapidity by which NDDs progress during their disease course represents one of the main challenges for many researchers. Generally, the RoP is indicative of a gradual and continuous decline of clinical symptoms and of a cognitive and/or motor impairment over time, which reflect the progressive loss of neurons and dysfunctions in brain neural networks [36]. For some NDDs, such as ALS, it is possible to numerically evaluate the RoP by using specific tests that assess disease severity and duration, but for other phenotypically more complex diseases, it is difficult to have a reproducible and validated severity parameter [37]. In particular, the observed RoP, described by Doody and colleagues [12], seems to give enough information about the disease progression in AD patients. Many factors could contribute to determining RoP in AD patients, such as the age at clinical onset, the initial severity of the symptoms, genetic predisposition, comorbidities, and environmental factors. Moreover, the role of biomarkers as predictors for RoP in different NDDs was widely investigated, showing their utility in clinical practice [38]. For this reason, we performed a retrospective observational study to investigate the potential prognostic role of CSF biomarkers at the time of diagnosis in predicting RoP in AD patients, regardless of their disease stage. 

We found no correlation between RoP and demographic and clinical features or APOE genotype of AD patients, suggesting that these features did not predict RoP in our population. The role of these features in predicting RoP is still a matter of debate, and results from earlier papers are not conclusive [39,40,41,42,43].

When we investigated the contribution of CSF biomarkers, we found a statistically significant positive correlation of RoP with pTau and NF-L and negative with Aβ 42/Ng ratio. These results are supported by previous findings showing that high CSF pTau concentrations were strongly associated with an increased risk of progression both from MCI to AD and from SCD to MCI [44,45,46], and our early study showed higher concentration of pTau in moderate AD than MCI [33]. Moreover, Aβ 42/Ng ratio also seems to be related to cognitive impairments [33,47] and was able to discriminate against MCI due to AD from moderate AD patients, proposing its role in the progression of synaptic pathology during the disease [33]. 

Stratifying patients according to the median value of RoP, we found that only CSF NF-L levels showed significant differences between groups and that patients with faster RoP showed higher NF-L levels than slow ones. When we investigated the role of CSF biomarkers in predicting the RoP, logistic regression analyses showed that only CSF NF-L levels at baseline predicted an increased risk for faster RoP. To do that, we identified the best NF-L cut-off with good sensitivity and specificity, able to discriminate lower from faster patients. Such cut-off, if validated in a larger cohort, can be considered as a predictor of RoP. These results support previous findings that reported the role of NF-L levels as potential prognostic biomarkers for different neurological disorders [27,32,48,49]. In particular, NF-L levels are able to predict survival and disease progression in ALS [17,50,51], PD [52], MS [21,53], FTLD [23], stroke [19,20] and sleep behavior disorders [54]. Moreover, they are able to discriminate not only the clinical conversion from MCI to dementia [22,23,30,55] but also from presymptomatic to symptomatic phase in individuals with familial AD [29].

In this context, the predominant role of neurodegenerative pathways emerged to support our findings. In AD, Neurodegeneration is a key event involved in cognitive decline and other clinical symptoms due to the neural loss in brain regions critical for memory and cognitive functions (such as the hippocampus and cerebral cortex), and NFs represent an indicator of neuronal damage [56]. Many mechanisms are involved in the extracellular release of NFs, such as (1) axonal degeneration due to the deposition of Aβ oligomers and pTau in brain parenchyma; (2) oxidative stress and mitochondrial dysfunctions associated with the production of Reactive Oxygen Species (ROS); microglial inflammation, which determine the release of inflammatory cytokines, proteases, and ROS. All these pathways are able to enhance neuronal damage, determining the extracellular release of NFs, including NF-L, detectable in CSF and blood [57].

One of the main limitations of our study is the small sample size (n = 56). Although this could affect the generalizability of the results, we adopted a methodological approach in descriptive analyses that considered the “effect size”, namely a measure of the strength of a relationship between variables able to show a better statistical power than the simple *p*-value. When we analyzed the differences between two AD subgroups stratified according to their RoP (Table 3), we found for NF-L an effect size of 0.455, which is considered a “medium-large” effect, demonstrating that the observed difference showed a statistical strength. Moreover, the use of logistic regression to analyze the relationship between biomarkers and outcomes allows for better management of confounding variables and the extraction of useful information even from smaller samples. Another limitation is the lack of longitudinal CSF biomarkers measurements and the assessment of their possible changes as the disease progressed. Without longitudinal data, the ability to predict future disease progression is limited. Repeated measures over time would allow for the development of more accurate predictive models, as trends in biomarker levels could be correlated with clinical outcomes such as cognitive decline or changes in disease severity. This could enhance the predictive power of biomarkers like NF-L and provide a more nuanced understanding of their role in AD. Furthermore, longitudinal studies can help establish temporal relationships between biomarker changes and clinical events. Finally, there is no independent validation of the results on another patient cohort, which could limit the confirmation of the validity of the identified biomarkers as predictors of disease progression. 

For these reasons, future perspectives are to extend our research to involve larger and more diverse patient cohorts, including the preclinical stage of the disease, to improve generalizability and statistical power, incorporating longitudinal biomarker monitoring, including blood-based ones, and providing deeper insights into disease progression. In particular, the study of a broader range of biomarkers, including those related to other aspects of AD pathology (e.g., inflammation, synaptic dysfunction), could reflect disease progression. Addressing confounding variables through detailed patient profiles and stratified analyses will enhance the accuracy of findings. Finally, the use of a multimodal approach that integrates CSF biomarker data with clinical and demographic features of AD patients, neuroimaging, and genetic information could improve the accuracy of prognostic models and enhance our understanding of AD RoP.

## 5. Conclusions

Predicting the single subject rate of progression in neurodegenerative disorders is a major challenge for clinicians. In our study, we investigated the prognostic role of CSF biomarkers and found a particular focus on NF-L in a cohort of AD patients. Our findings suggest that CSF NF-L at the time of diagnosis is a reliable predictor of RoP in AD, reinforcing its potential as a valuable prognostic biomarker. However, larger longitudinal studies are necessary to validate the prognostic utility of NF-L and to support its integration into personalized treatment strategies. Furthermore, stratifying patients based on their predicted rate of disease progression could be an effective strategy to improve future clinical trials. In conclusion, our results underscore the role of CSF NF-L as a biomarker for predicting disease progression in AD, emphasizing the importance of further research to confirm its clinical applicability and enhance patient care. 

## Figures and Tables

**Figure 1 brainsci-14-00960-f001:**
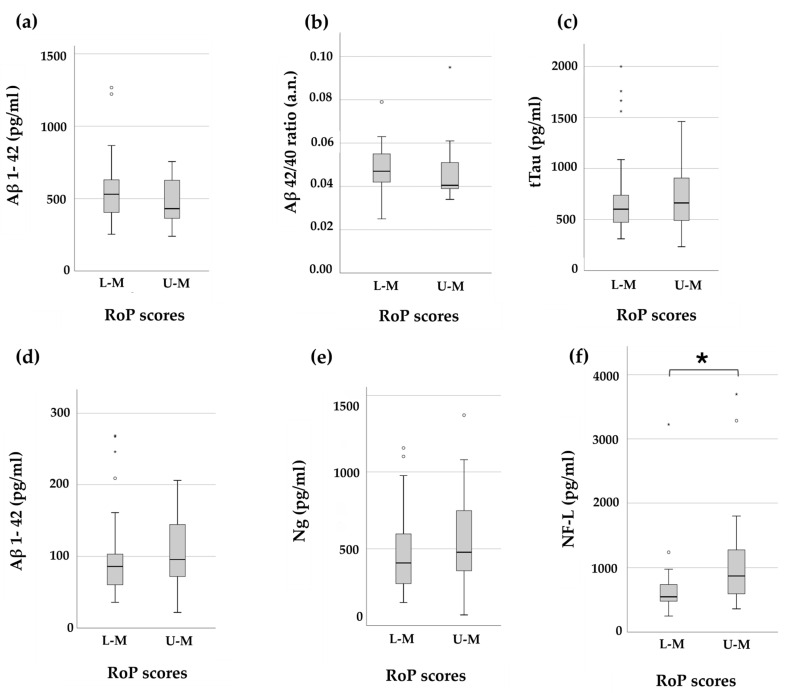
Box plots representing CSF levels of (**a**) Aβ 1–42, (**b**) Aβ 42/40 ratio, (**c**) tTau, (**d**) pTau, (**e**) Ng, and (**f**) NF-L in AD patients stratified on the basis of the median value of RoP (i.e., L-M and U-M). * indicates a statistical significance with a *p*-value < 0.05.

**Figure 2 brainsci-14-00960-f002:**
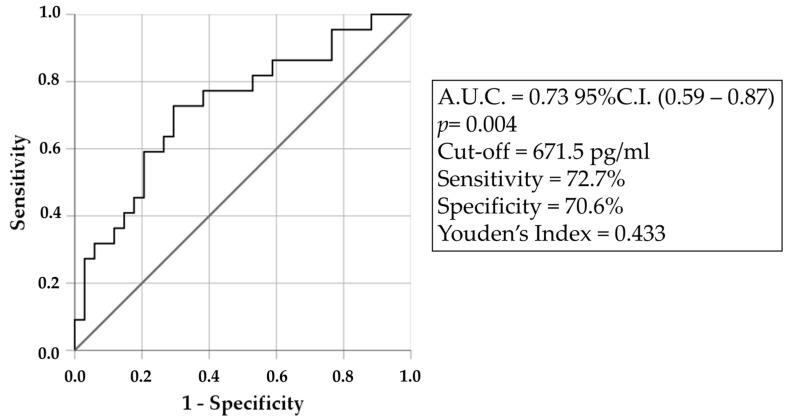
ROC curve analysis of NF-L, with cut-off value, sensitivity, and specificity, calculated according to Youden’s index.

**Table 1 brainsci-14-00960-t001:** Demographic and clinical characteristics of participants. Data are expressed as median with interquartile ranges (IQR). Mann–Whitney U-Test was used for continuous variables, while the Chi-square test was for categorical variables. ^A^ Lumbar Puncture; ^B^ Mini Mental Status Examination at time of diagnosis.

Variables	L-M (n = 34)	U-M (n = 22)	*p*
Age at onset (years)	69 (64–72)	67 (63–73)	0.906
Age at LP ^A^ (years)	73 (68–77)	75 (68–78)	0.775
Gender (M/F)	16/18	9/13	0.860
Education (years)	8 (5–13)	8 (5–13)	0.709
History of dementia (yes/no)	24/10	13/9	0.313
Diagnostic delay (years)	4 (3–6)	5(4–6)	0.334
Follow-up (years)	1.0 (0.9–2.0)	1.5 (1.0–2.0)	0.277
MMSE ^B^ (raw scores)	21 (17–27)	22 (16–24)	0.424
APO E ε4 (yes/no)	15/19	9/13	0.968
Memory onset (yes/no)	28/6	19/3	0.979

**Table 2 brainsci-14-00960-t002:** Relationship between RoP scores and CSF biomarkers of participants investigated by Spearman’s correlation analyses. Bold font indicates a statistical significance (*p* < 0.05).

Variables	Median Values (IQR)	*rho*	*p*
RoP (*a.n.*)	2 (−0.6–3.8)	-	-
Aβ 1–42 (pg/mL)	502 (382–627)	−0.140	0.303
Aβ 42/40 (*a.n.*)	0.045 (0.035–0.054)	−0.228	0.099
tTau (pg/mL)	619 (474–821)	0.220	0.103
pTau (pg/mL)	91 (61–128)	0.283	**0.034**
Ng (pg/mL)	438 (296–652)	0.248	0.087
Aβ 42/Ng (*a.n.*)	1.052 (0.799–1.609)	−0.293	**0.039**
NF-L (pg/mL)	617 (506–910)	0.432	**0.001**

*a.n.* = absolute number.

**Table 3 brainsci-14-00960-t003:** Biomarker levels in CSF of AD patients stratified on the basis of their RoP. Data are expressed as median with IQR. Mann–Whitney U-test was used to detect any difference between groups (i.e., L-M and U-M). Bold font indicates a statistical significance (*p* < 0.05).

Variables	L-M (n = 34)	U-M (n = 22)	Effect Size	*p*
Aβ 1–42 (pg/mL)	530 (399–631)	431 (363–627)	0.178	0.268
Aβ 42/40 (*a.n.*)	0.047 (0.042–0.055)	0.041 (0.039–0.059)	0.262	0.330
tTau (pg/mL)	602 (462–750)	663 (481–912)	0.156	0.326
pTau (pg/mL)	86 (59–103)	96 (69–149)	0.158	0.322
Ng (pg/mL)	369 (209–580)	458 (291–737)	0.193	0.257
Aβ 42/Ng (*a.n.*)	1.24 (0.70–1.67)	0.81 (0.060–1.43)	0.263	0.250
NF-L (pg/mL)	549 (456–751)	871 (577–1283)	0.455	**0.004**

*a.n*. = absolute number.

**Table 4 brainsci-14-00960-t004:** Binary-logistic regression analysis to investigate the predictive roles of CSF biomarkers in contributing to scores of RoP higher than the median value (U-M: >2). Bold font indicates a statistical significance (*p* < 0.05).

RoP (U-M)	B	s.e. ^1^	DF ^2^	OR (95%C.I.)	*p*
Aβ 1–42 (path.)	0.154	0.697	1	1.167 (0.298–4.571)	0.825
Aβ 42/40 (path.)	0.045	0.650	1	1.046 (0.293–3.738)	0.945
tTau (path.)	0.305	0.767	1	0.691 (0.302–6.103)	0.691
pTau (path.)	0.629	0.699	1	1.875 (0.506–6.954)	0.347
Ng (U-M)	0.674	0.587	1	1.962 (0.621–6.193)	0.251
Aβ 42/Ng (L-M)	1.025	0.599	1	2.786 (0.861–9.016)	0.087
NF-L (>cut-off value)	1.856	0.609	1	6.400 (1.940–21.112)	**0.002**

(*path.*): pathological values; ^1^ s.e. = standard error; ^2^ DF = degree of freedom.

**Table 5 brainsci-14-00960-t005:** Multivariate logistic regression analysis to investigate the contribute of different models to the prediction of faster RoP (U-M: >2). For each model, we considered only the values of the intercept. Bold font indicates a statistical significance (*p* < 0.05).

Model	RoP (U-M)	B	s.e. ^1^	*p*	AIC	BIC
Base	NF-L(>cut-off)	−1.386	0.456	**0.002**	11.11	15.16
1	Base + Ab 1–42 (path.)	−2.229	0.883	**0.012**	18.34	18.80
2	Base + Ab 42/40 (path.)	−2.293	0.866	**0.008**	15.90	22.04
3	Base + tTau (path.)	−1.176	0.784	0.134	16.85	22.95
4	Base + pTau (path.)	−1.251	0.641	0.051	17.17	23.25
5	Base + Apo E ε4	−1.274	0.508	**0.012**	17.45	23.53
6	Base + Ng (U-M)	−2.082	0.690	**0.003**	16.89	22.60
7	Base + Ab 42/Ng (L-M)	−2.202	0.692	**0.002**	16.75	22.49
8	Base + Ab 1–42 (path.) + Ab 42/40 (path.)	−2.554	0.996	**0.010**	25.06	33.16
9	Base + Ab 1–42 (path.) + Apo E ε4	−2.111	0.900	**0.019**	25.61	33.71
10	Base + Ab 1–42 (path.) + Ng (U-M)	−2.957	1.158	**0.011**	25.71	33.35
11	Base + Ab 1–42 (path.) + Ab 42/Ng (L-M)	−2.559	0.998	**0.010**	25.19	32.83
12	Base + Ab 42/40 (path.) + Apo E ε4	−2.333	0.881	**0.008**	24.07	32.17
13	Base + Ab 42/40 (path.) + Ng (U-M)	−2.488	0.968	**0.010**	22.31	29.97
14	Base + Ab 42/40 (path.) + Ab 42/Ng (L-M)	−2.124	0.921	**0.021**	20.43	28.08
15	Base + Apo E ε4 (yes) + Ng (U-M)	−1.969	0.701	**0.005**	24.31	31.96
16	Base + Apo E ε4 (yes) + Ab 42/Ng (L-M)	−2.104	0.720	**0.003**	27.10	34.75
17	Base + Ng (U-M) + Ab 42/40 (path.)	−2.287	0.743	**0.002**	27.49	35.14

(*path.*): pathological values; ^1^ s.e. = standard error. Akaike information criterion (AIC) and Bayesian information criterion (BIC) calculations are derived as follows: AIC = 2K − 2ln(L), where K = number of model parameters, and ln(L) = model log-likelihood. BIC = (RSS + log(n)d σ^2^/n, where RSS = residual sum of squares. n = Total observations. d = Number of predictors. σ^2^ = Estimate of variance of the error associated with each response measurement.

## Data Availability

The data presented in this study are available on request from the corresponding author. The data are not publicly available due to current privacy laws.

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
