# Peer review of "Cerebrospinal Fluid Neurofilaments Light-Chain Differentiate Patients Affected by Alzheimer’s Disease with Different Rate of Progression (RoP): A Preliminary Study"

_brainsci, 2024, doi:10.3390/brainsci14100960_

Round 1
Reviewer 1 Report
Comments and Suggestions for Authors
Title: Cerebrospinal fluid Neurofilaments light-chain differentiate 2 patients affected by Alzheimer’s disease with different rate of 3 progression (RoP): a preliminary study
Comments:
1. Is this study performed with the comparison of healthy individuals?
2. What is the normal range of CSF NF-L in the healthy individuals?
3. Is there any pathological changes occurs during the elevation of the biomarkers in AD patients?
4. Does this biomarker level have any correlation with age and gender?
5. Does this study proved preclinically as well?
Comments on the Quality of English Languagenil
Author Response
Please, see attachment

Reviewer 2 Report
Comments and Suggestions for Authors
In the manuscript entitled “Cerebrospinal fluid Neurofilaments light-chain differentiate patients affected by Alzheimer’s disease with different rate of progression (RoP): a
preliminary study”, the authors carry out a biomarker study to identify surrogates of disease progression in ADRD patients. Prior to acceptance for publication, the following points need to be addressed:
1. Line 40, “neuropathology began years…” should be “begins”
2. Line 73, NDD needs to be defined at first use.
3. Specify what assay was used for the ECLIAs. Who is the manufacturer, catalog number?
4. Specify what assay was used for the ELISAs. Who is the manufacturer, catalog number?
5. Add to Table 2 the median and the range for each of the variables/biomarkers measured as well as the values for the ROP scores.
6. Define a.n. in Table 3.
7. It would be beneficial to have a graphical representation in a box plot the data shown in Table 3, at least for NfL. Maybe this could be added to Figure 1.
8. In table 4, define cost and path?
9. Specify how was the best model chosen?
10. Consider fitting a multivariate linear regression to predict MMSE, in which case, are all variables needed to fit the best model possible? Consider using the AIC or BIC to come up with the best model? Evaluate the model for autocorrelation? Please, also write down the full predictive model as a formula. What would be the intercept of the model?
11. Why was a multivariate logistic regression model not considered?
12. In the discussion refer to the cut-off point of NfL that based on the study can be used as the predictor to progression.
Round 2
Reviewer 2 Report
Comments and Suggestions for Authors
My previous concerns have been satisfied.